# “We Absolutely Had the Impression That It Was Our Decision”—A Qualitative Study with Parents of Critically Ill Infants Who Participated in End-of-Life Decision Making

**DOI:** 10.3390/children10010046

**Published:** 2022-12-26

**Authors:** Maria Florentine Beyer, Katja Kuehlmeyer, Pezi Mang, Andreas W. Flemmer, Monika Führer, Georg Marckmann, Mirjam de Vos, Esther Sabine Schouten

**Affiliations:** 1Division Neonatology, Department of Pediatrics, Dr. von Hauner Children’s Hospital, LMU University Hospital, 80337 Munich, Germany; 2Institute of Ethics, History and Theory of Medicine, LMU Munich, 80336 Munich, Germany; 3Center for Pediatric Palliative Care, Department of Pediatrics, Dr. von Hauner Children’s Hospital, LMU University Hospital, 80364 Munich, Germany; 4Department of Paediatrics, Emma Children’s Hospital, Amsterdam University Medical Centre, 1081 HV Amsterdam, The Netherlands

**Keywords:** prematurity, parental involvement, withholding treatment (MeSH), shared decision making (MeSH), withdrawing treatment

## Abstract

Background: Guidelines recommend shared decision making (SDM) between neonatologists and parents when a decision has to be made about the continuation of life-sustaining treatment (LST). In a previous study, we found that neonatologists and parents at a German Level-III Neonatal Intensive Care Unit performed SDM to a variable but overall small extent. However, we do not know whether parents in Germany prefer an extent of more or sharing. Methods: We performed a qualitative interview study with parents who participated in our first study. We analyzed the semi-structured interviews with qualitative content analysis according to Kuckartz. Results: The participation in medical decision making (MDM) varied across cases. Overall, neonatologists and parents conducted SDM in most cases only to a small extent. All parents appreciated their experience independent of how much they were involved in MDM. The parents who experienced a small extent of sharing were glad that they were protected by neonatologists from having to decide, shielding them from a conflict of interest. The parents who experienced a large extent of sharing especially valued that they were able to fulfil their parental duties even if that meant partaking in a decision to forgo LST. Discussion: Other studies have also found a variety of possibilities for parents to partake in end-of-life decision making (EOL-DM). Our results suggest that parents do not have a uniform preference for one specific decision-making approach, but rather different parents appreciate their individual experience regardless of the model for DM. Conclusion: SDM is apparently not a one-size-fits-all approach. Instead, neonatologists and parents have to adapt the decision-making process to the parents’ individual needs and preferences for autonomy and protection. Therefore, SDM should not be prescribed as a uniform standard in medico-ethical guidelines, but rather as a flexible guidance for DM for critically ill patients in neonatology.

## 1. Introduction

Due to advances in neonatal intensive care, the survival rate of critically ill as well as extremely premature infants at the border of viability has significantly increased [1,2]. However, a considerable number of the surviving infants face an uncertain and unfavorable prognosis with a limited life expectancy and reduced quality of life [3,4]. Infants with a life-threatening condition and an extremely poor prognosis mostly die following a decision to redirect care from intensive to palliative care and to forgo (withhold or withdraw) life-sustaining treatment (LST) [5,6,7,8]. Such decisions require an ethical deliberation of norms and values. In decision making about redirection of care for an infant, the best interest of the child is of great importance. Yet, it is often difficult to determine to what extent the continuation of LST in cases with an uncertain or unfavorable prognosis is in a child’s best interest.

Medico-ethical guidelines not only recommend the involvement of parents in the decision making process in such cases but increasingly recommend shared decision making (SDM) as the preferred model of decision making [9,10,11,12,13,14]. There is no shared uniform definition of the concept of SDM [15]. SDM can be understood either as an umbrella term or as a narrowly defined concept. As an umbrella term, it refers to any form of collaboration between a medical team and parents [16]. As a narrow concept, it relates to a specific form of partnership between the two parties [17]. In this study, SDM is understood as a partnership that requires a communicative process that entails a joint deliberation with an exchange of values and preferences, resulting in a treatment decision upon which both parties agree [18,19,20].

Research on whether SDM should be implemented in end-of-life medical decision making (EOL-MDM) in neonatology has led to ambiguous results. Some researchers have argued that parental involvement in EOL-MDM may be unfavorable due to parents’ emotional involvement when coping with a serious health threat to their own child and feelings of guilt after a decision to forego LST [21,22]. The proponents of SDM assert that parents are the ones who are particularly affected by these decisions and therefore should have parental decision-making authority in such situations. Caeymaex et al. argued that participation in SDM could have a positive effect on the parents’ grieving process and pointed out that parents preferred SDM over models of EOL-MDM in which parents deliberate on treatment options alone [17,23].

Together with the heterogenous understanding of the concept of SDM, the various arguments concerning the (dis)advantages of parental involvement in EOL-MDM have led to different clinical practice models of SDM. Caeymaex et al. understand SDM as a process that entails a discussion on the nature of the decision, with an exchange of relevant medical information (including medically reasonable alternatives), followed by an exchange of family values and preferences that leads to a parental choice about the most appropriate decision [17]. On the other hand, Hendriks and colleagues describe their approach to SDM as follows: The medical team provides parents with comprehensive medical information to enable them to understand the best treatment option and give them the opportunity to ask clarifying questions. Next, the medical team recommends a treatment and asks the parents for their consent. This treatment recommendation is based on the medical facts, the infant’s best interests, and the parents’ values, which are determined without directly asking the parents [24]. Although both practice models claim to implement SDM, they lead to different modes of collaboration between parents and neonatologists. 

We could not identify a study that investigated whether and how SDM was practiced in German NICUs. Therefore, we previously conducted a prospective qualitative study in one of the level-III NICUs of LMU University Hospital, Munich, Germany [25]. In this study, we analyze whether SDM took place in conversations between parents and neonatologists. The conversations were part of the EOL-MDM process for infants who were born with life-threatening health conditions mostly due to extreme preterm birth, congenital malformations, or birth-related health impairments. We adapted a framework developed by de Vos et al. to determine whether SDM had taken place in these conversations [20]. Their framework of SDM consists of three stages (exchange of information, deliberation, reaching a decision), which may occur consecutively or iteratively. The extent of sharing (absent, minimal, moderate, great) in the EOL-MDM conversations was evaluated for all stages. We found a discrepancy between the prescription of SDM in the German guideline and its realization in neonatologist–parent conversations [25]. In the EOL-MDM conversations, SDM was only adopted to a small extent. The extent of sharing varied among cases, but overall, it decreased considerably over the stages of SDM. There was a low extent of sharing in deliberation and in the explicit conversation about how to reach a decision that both parties could agree upon compared to a high extent of sharing in the exchange of information. Neonatologists mostly used a communication strategy in which they suggested finding a decision together with parents, while at the same time seeking parents’ agreement to redirect care. 

To investigate how such realizations of SDM in EOL-MDM conversations are experienced by the involved parents, we conducted a follow-up interview study with the involved parents. With this study, we want to answer the question: How do parents perceive the realization of SDM in EOL-MDM for their severely ill or extremely preterm-born infant? We were especially interested in whether they perceived their often limited involvement in the decision making process as acceptable and beneficial. 

## 2. Materials and Methods

### 2.1. Study Context

This interview study is part of the ENFoLDING project (End-oF-Life Decision-making In NeonatoloGy), funded by the German Federal Ministry of Education and Research (grant number 01GY1718). The project aims to provide insight into pre- and postnatal EOL-MDM processes concerning (possible) NICU patients, with a special focus on the parental participation in this process. This paper reports data from interviews with parents who previously had been involved in a study investigating neonatologist–parent conversations by a qualitative content analysis [25]. The data originated from one level-III NICU and perinatal care center of the LMU University Hospital in Munich, Germany. 

Munich has approximately 1.5 million inhabitants. The LMU University Hospital, Grosshadern site, is one of three University hospitals in the city. In this NICU, approximately 600 newborns are treated per year, of which 1.5% die per year. The multi-professional team of this NICU (doctors, nurses, psychologists, chaplains) provides healthcare to extremely preterm-born infants, infants with congenital malformations, or birth-related health impairments.

### 2.2. Study Design

Since we wanted to gain insight into the perspective of parents on their participation in EOL-MDM, we chose a qualitative research design. Qualitative research is suitable for describing people’s subjective perspectives from their lifeworld [26]. The open approach of qualitative research gives participants more space to present their individual perspective than closed surveys [26]. In addition, it enables researchers to adjust their questions and research strategy more flexibly to the research situation, especially when participants show signs of distress or want to avoid certain issues. For a transparent description of the study method, this section is written according to the consolidated criteria for reporting qualitative health research (COREQ checklist). 

### 2.3. Methodology

We used semi-structured interviews to explore the perspective of parents with a lived experience of EOL-MDM. We applied qualitative content analysis (QCA) according to Kuckartz [27]. In our understanding, qualitative content analysis (QCA) is a method that involves systematizing communication to interpret its content. It enables identification of manifest and latent communication content. The systematization of relevant content for the research question is based on a category system. In this way, meaningful aspects of the data material can be identified. This allows interpretations that shed light on the research question. The method can be flexibly adapted to the research context [28].

### 2.4. Sampling and Recruitment

We selected a consecutive sampling method due to the limited number of potential study participants. The data acquisition period lasted from April 2019 to February 2021. All parents whose EOL-MDM conversations were included in the first study were eligible for interviewing [25]. Approximately three months after the death of a child, parents were invited to participate in the interview study. Parents whose child was discharged alive were contacted several weeks after discharge and invited to participate in the interview study. When giving their informed consent, parents were contacted by telephone to arrange the date for the interview. 

### 2.5. Data Collection

We conducted semi-structured interviews. The study entails the analysis of two sets of data: (1) transcripts of the qualitative interviews, (2) descriptive data from patient charts.

In the (1) semi-structured (group) interviews, parents were asked about their perception of their role in the EOL-MDM process. The interview grid (Box 1) was developed using the SPSS method designed by Helfferich [29]. The grid was developed based on the research interest and discussed within the research group. It was informed by literature about parents’ participation in EOL-MDM in NICUs [17,20,30]. The aims of the interview grid are shown in Box 1. The interview grid was pilot tested with one couple (case #4, Paul) and did not need to be adapted during the study process. This interview is included in the analysis. 

Box 1Interview grid
**Please tell us how you perceived the conversations with physicians during your time on the unit.**
Can you tell us about the conversation in which you learned that (child‘s name) was so sick that (he/she) might not survive? To what extent did you consider your part in those conversations appropriate—in terms of what you would have preferred at that moment?

The interviews were carried out once. According to the participants’ preferences, they took place either at home or at the hospital. Due to COVID-19 pandemic infection control measures, several interviews were conducted via a video conference tool. Interview participants were either one or both parents. The interviews were held by MB (clinical pediatrician/specialization in pediatric palliative care and post-doc). She introduced herself as neonatal fellow working in the NICU under a study and research association in the ENFoLDING project. MB was not previously involved in the medical treatment of the parents’ infants except in case #11, Peter. During the NICU stay of Peter, she worked as a resident on call, though she was not involved in the EOL-MDM conversations with the parents. She told participants that she wanted to investigate parents’ perspectives on EOL-MDM conversations. MB was trained in qualitative interviewing by KK (psychologist/bioethicist). KK has extensive experience in conducting qualitative interviews and in training early career researchers. 

Interviews were recorded on a portable audio-recorder, uploaded to a personal computer, and deleted from the recording device. Audio files were transcribed verbatim using simple transcription rules [31]. Transcripts were anonymized through the replacement of names, locations, and other identifiable information [32]. Participants were not actively invited to censor the transcripts and never requested this on their own.

Descriptive and demographic data on the infant (gestational age, sex, medical diagnosis, survival) and parents (age, education, profession, marital status and children, country of origin, native language, and religion) were collected from medical charts. 

### 2.6. Data Analysis

Qualitative content analysis (QCA) according to Kuckartz was chosen because it is suitable for the analysis of interview material and, beyond the analysis of manifest content, for the interpretation of respondents’ perspectives and hypothesis generation [27]. In this method, both deductive categories previously established based on the research question and inductive categories resulting from the collected data are used to analyze the interview material. QCA was performed deductive–inductively in an iterative process using the software MAXQDA (VERBI GmbH, Berlin, Germany). To analyze how parents experienced SDM in EOL-MDM in the NICU under study, the three stages of SDM after de Vos were applied to the analysis [20].

Consensual coding (MB and Kristina List (medical student)) according to Rädiker and Kuckartz was performed of 50% of the interview material [33]. All discrepancies were discussed and solved. In a research workshop with a group of qualitative researchers outside of the ENFoLDING project, preliminary results were discussed and revised. Explaining the study results to impartial researchers improved the validity, credibility, and hence quality of the data analysis. The coding guide is available from the authors by request. 

The researchers involved in the data acquisition and analysis self-reflected on the study question prior to the beginning of the research and did not advocate for or against SDM at that point. We performed a triangulation of perspectives within the interdisciplinary study group. Participants were not invited to provide feedback on the study results prior to its publication.

### 2.7. Research Ethics

The study was conducted in accordance with the Declaration of Helsinki [34]. Ethical approval was obtained by the institutional review board at the medical faculty of the LMU Munich (17-678). The study was registered in the German clinical trial register (DRKS00012671) before data acquisition started [35]. Prior to data collection, participants were informed verbally and in writing about the aim of the study and the intent to record the interview. Parents were informed about their voluntary participation, their right to withdraw and delete collected data at any time, and their option to skip interview questions or to answer off the record. The researchers were bound to medical confidentiality. Collected data were anonymized and stored securely according to current data protection regulations. All participants gave their written informed consent. As parents with a child who was or still is in a critical medical condition or as parents who are bereaved, participants were considered to be potentially vulnerable due to their life situation. The interviewer avoided questioning or raising doubts concerning the decision-making process. After the interview, parents were offered psychological or spiritual support by the psychosocial team of the NICU under study. All parents expressed feeling relieved after talking about their experiences, and none of them expressed feeling burdened by the interview itself. 

## 3. Results

### 3.1. Description of the Sample

We conducted eight interviews with 12 parents of eight NICU patients. We tried to include all parents who participated in the first study (23 parents of 12 children). The parents of two infants (case #2 and #8) could not be reached by telephone or mail. The parents of two infants (case #7 and #10) declined participation: one couple (case #7) did not want to look back on this time period but spontaneously said they had been satisfied with their experience, and the other couple (case #10) felt uncomfortable because of language barriers. The characteristics of the participants are listed in Table 1. Of the eight infants, five died after a decision to redirect care, and in three, the clinical situation stabilized. They were still alive at the time of the interview. 

### 3.2. Themes of the Qualitative Interviews

The themes of the qualitative interview analysis are summarized in Box 2. The parental experiences of their role in EOL-MDM are described according to the three stages of the SDM framework of Vos et al. The extent of sharing is described as small (parents described their involvement as minimal) or large (parents remembered taking part very actively or co-determining the decision). The parents evaluated their experienced role in the EOL-MDM positively or negatively for different reasons and reflected on their preference. 

Box 2Category system of the analyzed themes of the interview data
Parental experiences of their role in EOL-MDM
○Stage I: Exchange of information○Stage II: Deliberation○Stage III: Reaching a decision○Extent of sharingParental evaluation of their experienced role in EOL-MDM○Evaluation of parents who were shared DM to a small extent○Evaluation of parents who shared DM to a large extentParents’ preferences for their role in EOL-MDM


### 3.3. Parental Experiences of Their Role in EOL-MDM

In six cases, parents recollected going through all three stages of SDM (cases #4 Paul, #5 Max, #6 Anna, #9 Bella, #11 Peter, #12 Alex). In two cases, parents reported an exchange of information and then reaching a consensual decision with a neonatologist without a shared deliberation of values (cases #1 Katie and #3 Linda). 

Not all parts of the EOL-MDM took place in the recorded conversations. Some parents mentioned that parts of the process of EOL-MDM occurred outside of recorded conversations with the neonatologists. One father described a process of balancing values and reaching a joint decision in private: *“And following this conversation, we actually decided for ourselves, that we did not want any life-prolonging measures, that we did not want to let her suffer. […] then in the afternoon we had this decision-making conversation with the neonatologist, she took plenty of time for this conversation and we told her our decision.”* (case #6, Anna) 

In other cases, parts of the decision-making process took place with the extended family or at the patient’s bedside. 

### 3.4. Stage I: Exchange of Information

In all cases, parents remembered an extensive exchange of information in both directions (parents and neonatologists). Neonatologists particularly showed an interest in parents’ overall impressions of their infants at the beginning of conversations. Parents remembered receiving information about the infant’s condition on a large scope—foremost, an update about the medical situation including recent problems (deteriorations, complications). They were given treatment options with benefits and risks, and short- and long-term prognoses, including statistics on morbidity and mortality. Most of them remembered particular details about the nature, extent, and comprehensibility of the medical information. Later in the conversations, they were given opportunities to ask comprehension questions. One mother especially appreciated the time that doctors had taken and their attention to detail: *“We made a list with questions at home on beforehand, and when we asked her [the neonatologist], she took a lot of time and answered all of our questions, and she also told us when she didn’t know the answer for sure, I think that was good, she was honest, that was very important to us.”* (case #12, Alex) 

Parents appreciated when neonatologists communicated uncertainties that accompanied the decision. Receiving detailed medical information about their infant’s treatment was of great value to the parents. 

### 3.5. Stage II: Deliberation

In six (#4, #5, #6, #9, #11, #12) of eight cases, parents recollected deliberating health-related values with the neonatologist. Mostly, they remembered being asked what they wanted for their child. For them, the wellbeing of their child was of greatest importance. They expressed that they wanted their child not to suffer, e.g., not to be in pain, especially in situations of unfavorable prognosis. In one case, the father said: *“He asked us what we wanted for Max, I said: we do not want him to suffer, he should not be in pain, […] we [the parents] talked about that before we spoke to the neonatologists that afternoon, we just said… well no more life-prolonging measures when it is of no use anymore.”* (case #5, Max) 

Even when the options parents discussed with a neonatologist were related to mostly unfavorable outcomes (i.e., the death of the child), talking about these options made them feel somehow “in control” (#5, Max) and “involved” (#6, Anna) in the healthcare for their infant. They valued talking repeatedly with the neonatologist as the father of Max described: *“He [the neonatologist] recapitulated again and again, the arguments, our thoughts and words, that made it easier for us to accept, I mean to understand it, yes, so we could understand the options and we felt listened to. We had the impression, that we could talk about options, and that felt, or I mean it still feels, right to us.”* (case# 5, Max) 

For the parents of Max, talking to the neonatologist about options helped them to structure their thoughts; it supported their gaining insight into their own preferences. 

In the case of Anna, a newborn who suffered severe asphyxia during her birth, a decision was being sought. Her parents recollected discussing the impact of the expected disability on their daughter’s quality of life with the neonatologist. Anna’s father remembered the conversation as: *“Bottom line, the neonatologist said that disabled people often don’t perceive it [living with a disability] as that bad themselves […]”*. Then, the mother gave a different impression of the deliberation in Anna’s case: *“But we didn’t really talk about it like that, moreover we talked about that she would never be able to see, hear, taste, that is, her sensory organs would definitely never work […] That is where I simply said to the doctor, in my eyes, that is not a life worth living for, when she’s not able to take notice of anything in her surrounding that she can enjoy.”* (case #6, Anna) 

The decision was based on the doctor’s description of living with a severe brain injury. In his/her account, this meant that Anna would be permanently unable to interact with her environment. Her mother could not imagine how such a life could be enjoyable. The expected low quality of life provided a strong reason not to prolong LST any further, because the parents did not deem the intensive care measures in Anna’s best interest. 

### 3.6. Stage III: Reaching a Decision

In five cases (#4, #5, #6, #9, #12), parents remembered that a decision was being ascertained; in four cases (#4, #5, #6, #12), the decision to redirect care was evaluated, and in one case (#9), LST was continued. 

In three cases (#1, #3, and #11), the conversations referred to a possible deterioration of the infant’s condition in the future. The conversations were concerned with decision making in advance (or advanced care planning) in the event that the child’s condition would worsen. In the case of Linda (#3), an extremely preterm infant, her mother recalled receiving information on a severe complication without realizing that this complication might lead to an EOL-MDM situation in the future: *“This one conversation about the brain hemorrhage, […] I didn’t understand at the time, what this meant. Be it that I was still in shock or that it was only afterwards [that I understood the possible consequences of it].”* (case #3, Linda) 

After this conversation, Linda’s condition stabilized, the brain hemorrhage did not progress, and she did not develop any other life-threatening complications. After ten weeks of intensive care, Linda was discharged home. In the interview, Linda’s mother reflected that she would not have been able to decide on forgoing LST in her case due to the shock she was in. She felt that there was no other way than to completely trust the medical team, because mentally, she was not capable of taking part in such a decision-making process. 

The parents of three infants remembered the neonatologist stressing that “they did not have to decide” (#1, #3, #11), that it might even be impossible for parents to decide. The mother of Katie, an extreme premature baby that developed a severe brain hemorrhage, remembered the following from the conversation with the neonatologist: “(…) *and in that moment, well, we did not decide anything, and, I think, they [the medical team] did not expect us to do so, we were not expected to decide.”* (case #1, Katie) 

The parents did not consider it best for them to decide; they emphasized that the medical team would know best. For them, this was “neither positive nor negative” and “just the way it was”. LST was continued, and Katie’s situation did not further deteriorate. She stabilized and was discharged alive from the NICU after 14 weeks.

In two cases (#4 Paul, #12 Alex), the parents remembered that the decision to redirect care was proposed by the neonatologist as an option without an alternative. In the case of Alex, a newborn with severe congenital malformations whose situation deteriorated despite LST, his mother stated the following: *“And then, that evening they told us: ‘this [clinical deterioration] is what is going on and this [redirection of care] is what we will do’, and then they asked us if we agreed to that. They told us it was no use anymore, we could postpone it, but it would not make a difference for Alex, by then so much was clear to us.”* (case #12, Alex)

The neonatologist obtained consent to withdraw LST because of the futility of the situation. Alex’s parents understood the inevitability of this decision and felt that “everything possible had been tried”.

### 3.7. Extent of Sharing

Most parents (#4, #5, #6, #9, #11, #12) experienced a deliberation with a neonatologist during the EOL-MDM conversations, although the extent of sharing varied substantially in the participants’ reports. Parents reported a variation in the extent of sharing in the decision-making process. Most parents described their involvement as minimal (cases #1, #3, #4, #11, #12). Some remembered taking part very actively or co-determining the decision (cases #5, #6, #9). 

### 3.8. Sharing to a Small Extent

In five cases (#1 Katie, #3 Linda, #4 Paul, #11 Peter, #12 Alex), there was an overall small extent of sharing in the decision-making process. Parents recalled exchanging information extensively, followed by a decision being ascertained. Some parents did recall being asked about their values, but interpreted these questions not as an invitation to share in the deliberation. The mother of Alex (case #12), a newborn with congenital cardiac arrythmia, recalled that she was mainly told the medical facts and her child’s prognosis without feeling that she could participate in the decision-making process. She was asked questions, but she interpreted them as serving a different function in the conversation. She said: *“They asked us what we would wish for our son. And we just said, we wish that he is well, that he is healthy, that he does not have to suffer. That was it, nothing more was said […] So it seemed to us that they were preparing us for our son’s death.”* (case #12, Alex). 

Alex’s mother interpreted the doctor’s question about parental values as a rhetorical question. It was not perceived that the question was used to elicit the mother’s values to find a decision in an open-ended situation. The question was posed to convince the mother that the only thing that was left to want for her son was to relieve his suffering, because all the other things she had wanted were out of reach. 

### 3.9. Sharing to a Large Extent

In three cases (case #5 Max, #6 Anna, and #9 Bella), parents experienced a large extent of sharing in the decision-making process. The father of Bella, a premature infant with necrotizing enterocolitis, remembered extensively discussing his daughter’s medical situation and the treatment options (redirection of care or performing a surgery) with the neonatologist. Bella’s parents did not agree with the suggestion to redirect care and argued to give her daughter’s treatment another chance, despite the small odds of her survival. The father remembered: *“I decided that we should do the operation, he [the neonatologist] said, maybe it is best not do it. He said: ‘Then it goes as it goes and if she makes it, she makes it, if not then it is God’s will. Even if she gets the surgery, the chances of survival are only a few percent.’ […] Then I said: ‘Let’s do the operation, if she doesn’t survive the operation, then it is also God’s decision. If she survives, then God wants her to live’.”* (case #9, Bella) 

The father used this as an example to illustrate that in his opinion, the treatment preferences of both parties were jointly deliberated, and that both sides agreed upon the decision to continue intensive care and perform the operation. Bella was operated on but developed a severe septic shock with multiorgan failure. LST was withdrawn the following day. Both—father and neonatologist—referred to God as an authority that made the final decision. It was as if the father was hoping for a miracle, but at the same time, he was willing to do everything in his power to make that miracle possible. 

### 3.10. Evaluation of The Experienced Role in EOL-MDM

All parents said that their experienced role in EOL-MDM made sense to them now. For some parents, the experience was initially positive, and for others, it was negative, depending on the extent of sharing and their prior expectations. The reasons for a positive evaluation of a low extent of sharing were related to emotional distress, conflicts of interest, and trust in the medical competence of the medical team. The reasons for a positive evaluation of a large extent of sharing were related to aspects that empowered parents to be a parent, that enabled parents to protect their child from suffering and helped them to find closure. None of the parents who experienced a small extent of sharing evaluated this negatively. Prognostic uncertainty causing emotional distress was mentioned as negative by the parents of three cases: the mother of Peter (#11), who perceived a small extent of sharing, and the parents of Max (#5) and Bella (#9), who perceived a large extent of sharing. 

### 3.11. Evaluation of Parents Who Shared in DM to a Small Extent

Of those parents who experienced sharing to a small extent, some were relieved that they were protected by neonatologists from having to decide, by not leaving them with any more options. The father of Paul, a child who was born extremely premature, remembered: *“When I look back now: I would say: ‘That was the right decision’. That the doctor came in and said: ‘We should let him go, this is not going anywhere anymore, just like that, no choice.’ That was good, because if it was for us, we might have said: ‘No, we can’t stop’, he might have lived for a few more hours, or a day. But he would have suffered longer, and that would have been only for us, not for him.”* (case #4, Paul) 

By taking the decision away from the parents, the neonatologist in that case prevented prolonging the life of the child with LST for reasons related to the emotional connection between parents and their child. Parents were shielded from a conflict of interests: on the one hand, the wish to continue LST because they didn’t want to lose their child; on the other hand, the duty to protect their child from suffering. 

Another reason for parental satisfaction with a low extent of sharing was trust in the neonatologist’s professional competence. The mother of Paul, an extreme premature infant, recalled: *“I think that the doctor has so much professional experience that he knows when he says: ‘Your child shows us that it [the treatment] is of no use anymore’, or when he said ‘we can’t do anything anymore’, he doesn’t say that on a whim, or because another baby might need that NICU bed.”* (case #4, Paul) 

The neonatologist judged Paul’s situation as futile and sought parental agreement to redirect care. She agreed, as she trusted the neonatologist. She believed that the neonatologist, as an expert, would know what is best for Paul. She believed that the neonatologist’s core duty was to act in her child’s best interest. 

Some parents (cases #1 Katie, #3 Linda) described being relieved when not given the opportunity to decide. They said they would not have been capable of deciding, because they were in a state of extraordinary emotional distress. The mother of Linda, an extremely premature infant with severe intraventricular hemorrhage, explained it like this: *“I didn’t really realize it in the conversation […] I felt numbed. I think that it was better for me, personally, that I didn’t get it at that time. I think, with all the fears and these thoughts, it was only later on that I was better able to cope with them, I had regained some strength and handled it better.”* (case #3, Linda) 

The mother illustrated with this example that she did not see herself capable of deciding in that moment due to the state of shock she was in. She believed she probably would have lacked the mental strength to deal with having to decide. 

The mother of Peter initially felt insecure, because she lacked the medical knowledge to be able to participate in the decision-making. She would have preferred to be informed more extensively about Peter’s condition and the possible complications that could arise. *“I knew, I did not have to make a decision at that moment but by then I was very insecure. What will be next? What will I do when something happens? What are the options? What will I decide then?”* (case #11, Peter) 

Peter’s mother felt that she would have needed more information to take up a more active role in the advanced care planning. In that moment, she did not feel adequately informed, which made her feel helpless and overwhelmed. 

### 3.12. Evaluation of Parents Who Shared in DM to a Large Extent

In the case of Anna, a severely asphyxiated infant, the parents experienced SDM to a large extent and perceived this as adequate. Anna’s father remembered their initial evaluation: *“For us it was clear […] we wanted to have the machines switched off as soon as possible, because we just believed that she would not be in pain anymore and in a better place. We knew we were able to assess that really well. […] we already knew which way we wanted to go. It [the decision] was just right.”* (case #6, Anna)

Anna’s father explained their role in the decision making as a logical consequence of their parental responsibility to protect their child from suffering. As parents, they knew what was right for their daughter and considered it their duty to engage in the EOL-DM process.

Some parents (cases #5 Max and #6 Anna) voiced that they were satisfied with their role because it empowered them in their parental responsibility. The mother of Max, a newborn infant with fetal hydrops, recalled: *“We absolutely had the impression that it was our decision. Although, I don’t know for sure whether it was us or whether it was Max’s decision. But at least, we were able to decide for Max, so it seemed for us anyway. Or at least we had the impression: that was our decision to make, and this is the way we are going, that is ok.”* (case #5, Max)

The parents and the neonatologist agreed upon withdrawing LST for Max, as he was not showing any signs of improvement despite maximum intensive care. The parents of Max perceived deciding for Max as an act of parenthood. It was important for them that they could act on their parental responsibility. They perceived it as important that the neonatologist backed them up in their decision-making authority.

For the father of Bella, an extremely prematurely born infant with necrotizing enterocolitis, being able to join in the decision saved him from doubts of not having sufficiently advocated for his daughter. *“I [the father] think that’s great. Why should someone decide for me and for my child? I think that’s always great. […] Because in the end you ask yourself: Could something be better? Could something else have been done?”* (case #9, Bella)

Taking part in the EOL-MDM process served the father as a safeguard that all options were deliberated diligently. The thought that he had tried everything helped him to find closure. In moments of doubt about the legitimacy of the decision, the knowledge that he had ensured that no chance was left untaken gave him peace of mind. 

Other parents specifically valued the collaboration in the EOL-MDM process (case #5 Max, #6 Anna, #9 Bella). The father of Anna, who had suffered severe asphyxia, specifically appreciated the collaborative relationship with the neonatologist in the decision-making process. He described the process as follows: *“….. you try to solve the problem together. That’ s how it feels to us now, that we’ve been working on it together. That they tried to get us on board and […] not making a decision about the parents, but with the parents.”* (case #6, Anna) 

For the parents of Anna, being able to join the neonatologist in the DM process alleviated the feeling of complete helplessness. 

Some parents (case #5 Max, #9 Bella) who retrospectively valued the large extent of sharing for various reasons as positive also recalled initial negative feelings. They described an overload of emotions such as anxiety, insecurity, and helplessness. This was especially expressed in situations in which the parents’ agreement with a suggestion to forgo LST was sought. Parents felt insecure with the uncertainty that accompanied their child’s condition and prognosis. The mother of Max explained her feelings: *“The fact was, that not much was known about the disease, I mean, Max’s condition was so unclear. It was clear it was not okay, but why they [medical team] did not know. It was clear it was bad, but still this uncertainty stressed us. In the end we [the parents and the neonatologists] sat together and we had to decide. In that moment we were so tense and felt overwhelmed.”* (case #5, Max) 

As Max’s parents had a medical background, they could understand that there would always be some prognostic uncertainty, as Max had a rare condition, and his underlying disease was unclear. Still, the fact that there was prognostic uncertainty bothered them in the EOL-MDM situation. The decision-making responsibility strained them in that moment, but looking back, the “situation made sense” for the parents.

### 3.13. Parents’ Preferences for Their Role in EOL-MDM

For all included parents, taking part in EOL-MDM about their seriously ill infant in the NICU was a unique experience. Parents were unlikely to face such a decision again in their lifetime. This unique situation made it hard for parents to imagine that things could have been done differently than the way they experienced them. This becomes apparent in the case of Peter, an extremely premature born infant. His mother answered the question about how she felt about her participation in the DM conversation as follows: *“Well, I think I would not have wished for anything else, also not concerning my involvement. It was an exceptional situation, I don’t think one could have done things much differently anyway.”* (case #11, Peter)

Peter’s mother did not feel as though there were alternative options to choose from due to the unfavorable prognosis. She reckoned the clinical situation would take its course, and there was not really anything that could change that. This feeling made her accept her role, as she felt nothing else could be done. In her opinion, no one could have changed the outcome for Peter, so it would not have made a difference in the end whether she had experienced a different level of involvement in the decision-making process. 

Some parents said that they did not have the expertise to be more involved. The mother of Alex, an infant with severe congenital malformations, reacted to the question of whether she preferred a different experience concerning her participation as follows: *“No, not really. Because we didn’t know any better. You have to believe everything that you are told. We are not the doctors”* (case #12, Alex)

She reflected that as a parent, she felt overcome by the situation. Alternative options in that moment would not have made a difference for her, because in her opinion, as a parent, one lacks the medical expertise to decide. 

In one case, a mother indicated that she might have needed more information in order to participate more actively in the decision-making process. This could have helped her to become aware of her values and treatment preferences. The mother of Peter, an extremely premature born infant, explained it like this: *“I think it would have been better if I had known about all the possible complications, so I could have let it cross my mind. Because I am such a person, I always want to know everything. If it [a particular complication] does not occur, it’s fine, if it does, then I’m prepared for it. […] And then I know what to say when they ask me: What would you prefer in this or that situation? What do you want us [doctors] to do? What about the machines, what is it that do you do not want? I think that would have been good.”* (case #11, Peter)

Without the possibility to think about it in advance, the conversation took her off guard. She did not join in the DM process in the way she could have when the neonatologist had talked her through all the possible scenarios. 

When being asked about their preferred participation, parents instead frequently referred to their overall positive experience in the NICU. Most parents explicitly stated that they did not want to go somewhere else in the case of a similar situation in the future. 

## 4. Discussion

This study sheds light on the perspectives of parents who experienced SDM to different extents within the intensive care for their critically ill newborn infant. To our knowledge, this is the first study in which German parents were interviewed about their experience with EOL-MDM. There is a remarkable correspondence between our analysis of the physician–parent conversations and the parents’ perceptions, which is rarely found in similar studies [25,36]. The results show that parents valued their involvement in the EOL-MDM as beneficial and acceptable for different reasons regardless of the extent of sharing they experienced in DM. All parents appreciated being extensively informed about the medical situation und specifically valued being prompted to ask questions. For some parents, more involvement in decision making would have caused emotional distress and conflicts of interest they preferred to avoid. Others felt empowered in their parental role by engaging in EOL-MDM, and it helped them to find closure afterwards. Through elaborating how parents experienced their participation, we have contributed to knowledge on parents’ adequate role in EOL-MDM for their critically ill newborn infants. 

In the German context, there is a dissent among healthcare professionals as to whether parents should be involved more and hence burdened with decision-making responsibility. Medical guidelines in North America and Europe recommend SDM for EOL-MDM in neonatology, and since 2014, the German guidelines on healthcare for preterm born infants at the border of viability have integrated SDM into their guidance [12]. One of the reasons for integrating SDM were the results of a study by Caeymaex et al. This study showed that parents preferred SDM over other forms of decision making (more precisely, informed parental DM). SDM is understood heterogeneously in the research literature, which makes comparisons of parents’ preferences for SDM between different study contexts challenging.

In neonatology, EOL-MDM entails balancing norms and values. SDM integrates the values of both parties involved, the values of parents and the professional values of the medical team. Neonatologists use the ethical principles of beneficence and non-maleficence in order to act in the best interest of the child, and at the same time, they aim to respect parental authority. As children differ with regard to the prognosis of intensive care treatment, and parents have different arguments for their preferred role with regard to EOL-MDM, the outcome of this balancing of obligations can differ from case to case. There is not only one way to act morally. Feelings of guilt, coping with bereavement, and finding closure are key elements in the arguments for the parental preference for either more or less involvement. 

In general, parents have a right to participate in EOL-MDM. Some even would go so far as to say they had a moral obligation to make decisions in their child’s best interests in all circumstances of life [37]. Therefore, in EOL-MDM situations, parents are called on to participate in decision making on the grounds of their parental authority. In addition, they are the ones most affected by the consequences of a decision (not) to forgo LST. In light of these arguments, the viewpoints of both types of parents, those who appreciate SDM to a large extent, as well as those parents who appreciated less involvement, are comprehensible. Not having to have a say in such a grave decision was appreciated by parents, as it protected them from later feelings of guilt. They did not perceive a responsibility for the death of their child in cases in which a decision to redirect care from intensive to palliative care was taken. On the other hand, other parents saw it as their moral obligation and their fiduciary duty to participate in the EOL-MDM for their infant. It increased their capacity to cope with their loss. Due to the intensive care setting, with its technical features, parents might feel especially restricted in their ability to exercise their parental duties. In this way, they at least could act out this parental responsibility for their child. 

Overall, the parents in this study valued their involvement in the EOL-MDM as beneficial and acceptable for different reasons. This has also been seen in other interview studies concerning EOL-MDM with parents: most of them were satisfied with the role they took on in the decision making, though they experienced different levels of involvement in the decision-making process [24,38,39]. This could reflect parents’ true preferences, based on their self-understanding and their moral intuitions. Another possible explanation for parents to find their involvement beneficial independent of the extent of sharing could be sought in the way people cope with traumatic experiences. Being able to integrate the traumatic experience of losing a child in a biographic narrative and to give meaning to this may help parents to restore their perspective of the world as something that can be controlled and seems comprehensible [40]. It can be seen as part of an effort to try to make sense of the situation in order to find closure [41]. This could have led to the evaluation that the experience as a whole was meaningful, and therefore, their involvement was acceptable.

In the NICU under study, different parents experienced different realizations of SDM, identified as different extents of sharing in DM. On a continuum between a paternalistic approach and an informative approach, the approach in this NICU can be positioned in between these two extremes, but with a considerable variability between single cases. Different realizations of SDM within one NICU are common. Studies from other countries also describe that parents experience different levels of involvement in EOL-MDM. A French study identified three types of decision making strategies: medical decision making, shared decision making, and informed parental decision making [17]. In a study from Scotland, parents perceived that in EOL-MDM, the decision making authority could be with the medical team, with the parents, or upheld by a joint decision [42]. These other conceptualizations make clear that it is challenging to distinguish SDM from other forms of collaborative MDM. 

Our study confirmed that different parents have different preferences regarding their involvement in EOL-MDM. Brinchmann and colleagues showed that parents wanted to participate in decisions but not to decide. They found that parents did not want to have the final word in decisions concerning their infants’ future life or death [16]. McHaffie et al. showed that the majority of the parents of critically ill newborn infants preferred an active role in the EOL-MDM [43]. Caeymaex and colleagues reported that many parents appreciated it when they could voice their opinion in EOL-MDM. Although they concluded that parents explicitly preferred to share the responsibility in decision making, they also pointed out that participation preferences of parents can vary. They argued that SDM should serve parents with the opportunity to decide for themselves the role they want to play in the decision making [17]. The variability of the results in these studies underpins that there is not one strategy for EOL-MDM that fits all. 

The approach to SDM that was used in this NICU falls short on a couple of aspects. In our understanding of SDM, it requires a communication process between parents and neonatologists that entails a joint deliberation with an exchange of values and preferences that should be concluded by a decision upon which both parties agree upon [18,19,20]. Since in the exemplary NICU under study, neonatologists and parents approached EOL-MDM differently in different cases, it is hard to explain this variability [25]. Overall, neonatologists and parents conducted SDM in most cases only to a small extent. Neonatologists tended to offer parents a choice, while at the same time, they tried to convince them to consent with their suggestion to redirect care from intensive to palliative care for their infant in a critical health condition. Another important aspect of SDM that was only partially realized is its key aspect: a shared deliberation. We understand shared deliberation as a dialogue between parents and neonatologists in which they actively discuss and balance values and preferences in order to reach a bilateral understanding and agreement on a preferable course of action. Only the parents of Anna described such a process. Other parents recollected that the neonatologist merely asked them what they wanted for their child. This mainly evolved around aspects of health-related beneficence, while neglecting other aspects of quality of life. Another important key element for SDM was rarely recalled by parents: reaching a decision upon which both parties agreed. In those cases in which a decision was reached, most parents remembered that a decision was ascertained and their consent obtained. This approach seems to align with an “ethical model” for EOL-MDM described by Hendriks et al. rather than with SDM based on a shared deliberation [24,44]. In three cases, the aspects parents remembered from the conversations showed more resemblance with breaking bad news conversations according to the SPIKES Framework (Setting, Perception, Information, Knowledge, Empathy, Summarize) [45]. The parents of Alex recalled the neonatologist asking them to summarize what they had understood of the clinical situation of Alex (P). This was followed by the message that the decision to redirect care was without an alternative (I). They valued that the neonatologist showed empathy (E) and described the following steps (S). Although Alex’s parents appreciated their role in the DM process, it can be questioned whether it was actually SDM that they experienced. The same holds true for the parents of Katie; they did not consider it their place to decide, and they emphasized that the medical team knew best. They perceived no joint deliberation nor a shared decisional authority. 

A possible explanation as to why SDM is not realized in EOL-MDM in this NICU may be that neonatologists do not know how to perform SDM. The German medical school system provides training in breaking bad news conversations but offers no mandatory training in how to perform SDM with patients. Further, pediatric residency and neonatal fellowships in Germany do not cover this sort of training. A different explanation could be that the practitioners’ moral intuitions not to prolong intensive care treatment in cases with a poor prognosis are in conflict with the medico-ethical guidelines to perform SDM, which results in an impossible compromise of attempting to do both at the same time. In order to be able to better distinguish different approaches to collaborative EOL-MDM, we are currently conducting a meta-ethnography of comparable qualitative studies [46].

Although parents appreciated their experience overall, some parents initially evaluated their involvement negatively. Yet, a difference between an initial and a retrospective evaluation can be observed: in the end, most of the parents found their participation beneficial regardless of the extent of sharing they had experienced. A possible explanation for this may be that the neonatologists adapted and hence “tailored” their EOL-MDM approach according to assumed or expressed parental preferences. Mercuri et al. stressed the importance of a tailored approach in their recent contribution on why guidelines should not include one specific decision-making strategy such as SDM. They propose that the suitable decision-making strategy is always an individual and context-sensitive issue that should be realized by the parties involved instead of a single standard approach recommended by specific guidelines [47]. A preference-tailored approach is successful under the premise that neonatologists are either right in their assessment of parental preferences for participation, or that parents are given enough opportunities to actively advocate for more or less involvement and re-negotiate their role. A disadvantage of relying on parents to advocate for more or less participation by themselves is that not all parents may feel empowered enough to claim their preferred degree of participation. Parents may not have the communication skills or the strength to advocate for a different approach during such a stressful life event. This may amplify existing social inequalities. Apart from tailoring to parental preferences, there should be room for tailoring to the child’s condition, which might point in the direction of redirection of care because of a futile treatment approach. 

Consequently, we recommend that guidelines such as the German guideline for neonatal care in cases of extreme prematurity allow for a case-based determination of an adequate decision making approach instead of advocating for SDM as the only recommended practice of EOL-MDM. Future research should explore in detail the variety of approaches to collaborative EOL-MDM in neonatology and the justifications for its application in concrete cases.

### Limitations

There are several limitations to our study. In this study, parents from only one NICU in Germany participated. It is a typical NICU of the highest level, but we do not know whether the results of this study can be transferred to other centers as such [48]. This study is prone to selection bias in its sampling. Parents from four out of twelve eligible cases were not willing to participate. It could be that these parents were neutral, critical, dissatisfied, or traumatized and therefore did not take part in an interview. On the other hand, it could be that parents who were willing to participate in an interview study that was initiated by the NICU in which their child was treated were biased to evaluate their experience as positive. 

The interviewer (MB) worked as a physician in the NICU under study, but she was not directly involved in the clinical care of the infants apart from case #11. Nevertheless, this could have caused parents to act reserved in openly (on record) expressing worries or dissatisfaction about their experience. There was a time latency of at least three months between the EOL-MDM situation and the interview. In light of the fact that it was an existential experience, it may be assumed that despite the time that had passed, parents still remembered it well. However, in order to cope with the experience, parents could have altered or forgotten memories of their experiences. All bereaved parents had a follow-up meeting with the head of the department before being invited to participate in the interview. This follow-up meeting is standard care and offers parents the possibility to ask questions and discuss their NICU experience. Talking with the head of department before the interview could have changed their minds about their experience and consequently their evaluation. Furthermore, recall bias was brought up spontaneously by parents. Emotional distress, shock and, in the case of the mother, her own compromised physical state, were mentioned as reasons for parents to have forgotten details of the conversations. At the same time, parents showed an extraordinary ability to recall detailed information of their experience, sometimes even literally recalling wording and phrases from the conversations with the neonatologists. 

When invited for the interview, parents were informed about the purpose of the study, but not that the focus was their participation in the DM process. This had an impact on the scope of our data. In order not to negatively impact the psychological processing of the experience or to arouse feelings of guilt, regret, or doubt, questions about the participation of the parents and their satisfaction were asked sensitively. Parents were not informed about alternatives, such as the possibility of more or less participation in the DM process in other cases. 

The researchers’ subjective perspectives and interpretations of the data plays a significant role in qualitative studies. To enhance intersubjective comprehensibility of the study’s results, half of the interview material was analyzed by two researchers individually and coded consensually. Furthermore, the study’s major results were discussed within the research team and with impartial researchers in the context of a qualitative research workshop. Furthermore, we added the following paragraph to the limitations: future studies should investigate decision making in various centers to improve the generalizability of the results. It would also be advisable to separate the professional role in patient care (as a neonatal fellow in this study) from the professional role in research (as an interviewer). Furthermore, a longitudinal study could better account for contextual factors that influence the way parents evaluate their experience.

## 5. Conclusions

In order to implement SDM in EOL-MDM, both parents and neonatologists have to be willing to share in the process. In this study, the extent of sharing between parents and neonatologists in EOL-MDM differed from case to case, but mostly, SDM was only conducted to a small extent. The decision-making process was always appreciated and found beneficial. Therefore, it cannot be assumed that all parents prefer the same extent of SDM; thus, whether SDM is the appropriate decision-making model needs to be decided individually. A preference-tailored and context-dependent approach appears to be more suitable than a uniform recommendation of SDM as the favored approach for EOL-MDM for critically ill newborn infants.

## Figures and Tables

**Table 1 children-10-00046-t001:** Description of sample.

	Information about the Child	Information about the Parents
Case Nr.	Name	Main Diagnosis	Course	First Language	Interview Presence
1	Katie	extreme prematurity	Discharged alive from NICU	German	♀
3	Linda	extreme prematurity	Discharged alive from NICU	German	♀ + ♂
4	Paul 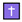	extreme prematurity	Withdrawn LST	German	♀ + ♂
5	Max 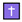	acquired disease	Withdrawn LST	German	♀ + ♂
6	Anna 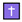	acquired disease	Withdrawn LST	German	♀ + ♂
9	Bella 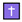	extreme prematurity	Death despite LST	Other	♂
11	Peter	extreme prematurity	Discharged alive from NICU	German	♀
12	Alex 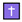	severe cong. malformations	Withdrawn LST	German	♀

Legend: Case 2 and 8 are missing due to loss to follow-up, case 7 and 10 declined participation, names are anonymized, 
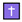
 deceased, LST life-sustaining therapy, ♀ mother, ♂ father.

## Data Availability

Restrictions apply to the availability of these data because there are ethical and privacy issues present.

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
