# Peer review of "“We Absolutely Had the Impression That It Was Our Decision”—A Qualitative Study with Parents of Critically Ill Infants Who Participated in End-of-Life Decision Making"

_children, 2022, doi:10.3390/children10010046_

Round 1
Reviewer 1 Report
This is an interesting and original qualitative study of the experience of end of life decision making for eight parents of critically ill infants from a single tertiary neonatal centre in Germany. The study provides novel and valuable insights into the experiences of parents at a single neonatal centre. The value of the study is limited to some extent by the small numbers and single centre basis but it still makes a valuable contribution. The methodology is well-described and the findings are discussed appropriately and in detail. The references are appropriate. Three minor comments:
1. The finding that shared decision making (SDM) was in reality carried out only to a small extent (paragraph beginning line 726) seems to me to be a very important take-away for the paper as a whole. In my view this finding should be summarised in the Abstract and also in the Conclusion section.
2. I see in the legend to Table 1 that the names are anonymised. This should be stated clearly in the Methods section to avoid confusion.
3. It would be helpful if the researchers could make some recommendations for future studies of SDM in neonatal practice. What have the researchers learnt about methodology which will be of help to future researchers in this area?
Author Response
Dear Reviewer,
thank you very much for the review.
We made the following changes according to your comments:
1.The finding that shared decision making (SDM) was in reality carried out only to a small extent (paragraph beginning line 726) seems to me to be a very important take-away for the paper as a whole. In my view this finding should be summarised in the Abstract and also in the Conclusion section.
Thank you very much for this suggestion. We edited the abstract and the conclusion (highlighted in yellow) according to your suggestion. Abstract (line 22) ‘Overall, neonatologists and parents conducted SDM in most cases only to a small extent.’ Conclusion (line 837) ‘In order to implement SDM in EOL-MDM both parents and neonatologists have to be willing to share in the process. In this study, the extent of sharing between parents and neonatologists in EOL-MDM differed from case to case, but mostly SDM was only conducted to a small extent.’
2.I see in the legend to Table 1 that the names are anonymised. This should be stated clearly in the Methods section to avoid confusion.
Thank you for this important remark. We added a statement concerning the anonymization in the methods section (line 208). ‘The cases were anonymized by using pseudonyms for the infants.’
- It would be helpful if the researchers could make some recommendations for future studies of SDM in neonatal practice. What have the researchers learnt about methodology which will be of help to future researchers in this area?
We added the following sentence to line 780 in the discussion: ‘Future research should explore in more detail the variety of approaches to collaborative EOL-MDM in neonatology and the justifications for its application in concrete cases.’. Furthermore, we added the following paragraph to the limitations: ‘Future studies should investigate decision-making in various centers to improve the generalisability of the results. It would also be advisable to separate the professional role in patient care (as a neonatal fellow in this study) from the professional role in research (as an interviewer). Furthermore, a longitudinal study could better account for contextual factors that influence the way parents evaluate their experience.’
Reviewer 2 Report
The authors undertake a study of the extent and preference of parents for shared medical decision-making in the NICU. The semi-structured interviews and qualitative content analysis are an effective approach for this topic. This reviewer found several areas that could be consolidated and clarified to ensure that the readers are able to fully understand the findings.
1. The insertion of quotes in the text makes it unwieldy to read. These would be better categorized in a table by codes utilized.
2. Including the case # and name is also distracting to the reader and makes assimilation of the findings and thus the conclusion less strong. This reviewer suggests removing those from the text of the manuscript.
3. Each section would benefit from shortening and consolidation of descriptions. The length of the paper diminishes its effectiveness, findings and implications.
4. That the interviewer avoided intimating doubt of parental decision-making is imperative and an important aspect of the ethics of the interviews and study. It would be helpful to know what coaching was given to ensure that this was able to be avoided by the interviewer.
5. Withdrawal of LST is used in the manuscript. The more accepted term is moving to "redirection of care" and it might benefit the manuscript to alter the language.
6. One of the interviewers is described as a "neonatal fellow working in the NICU under study." Please clarify whether this fellow was involved in the care of the infants and their parents. Fellows often have interaction and participate in the care of infants in all areas of an NICU and thus might skew the results of the interview.
7. The title of the manuscript is confusing to this reviewer. It indicates that parents uniformly felt that that "it was (their) decision" and yet your study conclusions and findings do not substantiate this.
Author Response
1.The insertion of quotes in the text makes it unwieldy to read. These would be better categorized in a table by codes utilized.
Thank you for this remark. Qualitative research relies on quotes or extracts from interviews, documents or field notes to illustrate the analysis. Quotes should be tagged in the text with appropriate identifiers. Enough quotes need to be included in the text to enable the reader to immediately judge the credibility of the interpretations.[1] Each quote was introduced in order to provide the reader with context information, the quote itself provides the reader with the information that is used for the interpretation that follows the quote.
2.Including the case # and name is also distracting to the reader and makes assimilation of the findings and thus the conclusion less strong. This reviewer suggests removing those from the text of the manuscript.
Thank you for this suggestion. We reported our findings according to the COREQ (COnso- lidated criteria for REporting Qualitative research) guidelines, which is a 32-item checklist.[2] The checklist recommends that each quotation should be identified by participant number. This enables the reader to identify whether there is consistency between the data presented and the findings. It enhances transparency and trustworthiness to the findings and the interpretation of the data.[3]
Furthermore, it enables the reader to identify that quotations from different participants were used, instead extracting of all findings from one or two interviews. Besides using case numbers, we deliberately used pseudonyms to keep the humanness. The data concern severely ill newborn infants and it was of great importance to us preserve this human factor.
3.Each section would benefit from shortening and consolidation of descriptions. The length of the paper diminishes its effectiveness, findings and implications.
Thank you very much for this comment. We shorted the methods section and made an effort to consolidate the description of each single theme in the results section.(see track changes) Unfortunately, we could not remove greater parts or themes from the results section without disrupting the line of argument. The data of the interviews offered many important themes, every theme needs support of parental quotes. These quotes are important to give the parents a voice in the scientific discourse. General consolidation of the themes would not pay enough respect to the importance and the nuances of the various different parental perspectives. As the journal has no maximum word count, we would like to keep the comprehensive portrait of the parental perspective.
4.That the interviewer avoided intimating doubt of parental decision-making is imperative and an important aspect of the ethics of the interviews and study. It would be helpful to know what coaching was given to ensure that this was able to be avoided by the interviewer.
Thank you for this important remark. It was the interviewer’s own conviction to avoid parents to doubt the decision or their participation in the decision-making process. She performed a very cautious and reticent way of interviewing. The interviewer evaluated the interviews with the research team on a regular basis and the research team reflected on her interviewing manner.
- Withdrawal of LST is used in the manuscript. The more accepted term is moving to "redirection of care" and it might benefit the manuscript to alter the language.
Thank you very much for this comment. We altered the language through the manuscript were we thought it was fit (highlighted in yellow).
- 6. One of the interviewers is described as a "neonatal fellow working in the NICU under study." Please clarify whether this fellow was involved in the care of the infants and their parents. Fellows often have interaction and participate in the care of infants in all areas of an NICU and thus might skew the results of the interview.
We commented on this in line 187 in the methods section ‘MB was not previously involved in the medical treatment of the parents’ infants except in case #11, Peter. During the NICU stay of Peter she worked as a resident on call, though she was not involved in the EOL-MDM conversations with the parents.’ as well as in line 806 in the limitations section ‘The interviewer (MB) worked as a physician in the NICU under study but she was not directly involved in the clinical care of the infants apart for case #11. Nevertheless, this could have caused parents to act reserved in openly (on record) expressing worries or dissatisfaction about their experience.’ (both sections are highlighted in green)
- The title of the manuscript is confusing to this reviewer. It indicates that parents uniformly felt that that "it was (their) decision" and yet your study conclusions and findings do not substantiate this.
Thank you for this comment. It is customary in qualitative research to use a single in vivo quote as a title. By marking it as a quote it reflects the perspective of one person and does not aim to offer a general explanation for all parents. We edited the title (deleting the exclamation mark) to match it to the original quote in the main text. Furthermore, we edited the explanatory paragraph following this quote in order to underline our interpretation of the quote.
‘"We absolutely had the impression that it was our decision. Although, I don't know for sure whether it was us or whether it was Max’s decision. But at least, we were able to decide for Max, so it seemed for us anyway. Or at least we had the impression: that was our decision to make, and this is the way we are going, that is ok." (case #5, Max) The quote illustrates the different nuances of perceiving decision-making authority and the meaning that parents attribute to their perceived role in decision-making.’
We identified these different nuances in many reports of parents in this study. The quote shows that parental perceived experiences are highly complex and individual. This is in line with our conclusion that a preference-tailored and context-depended approach would be more suitable than SDM for all parents.
References:
- Green, J.T., N. Qualitative methods for health research; SAGE: 2018.
- Tong, A.; Sainsbury, P.; Craig, J. Consolidated criteria for reporting qualitative research (COREQ): a 32-item checklist for interviews and focus groups. Int J Qual Health Care 2007, 19, 349-357, doi:10.1093/intqhc/mzm042.
- Cote, L.; Turgeon, J. Appraising qualitative research articles in medicine and medical education. Med Teach 2005, 27, 71-75, doi:10.1080/01421590400016308.